# Enhancing Self-Esteem and Body Image of Breast Cancer Women through Interventions: A Systematic Review

**DOI:** 10.3390/ijerph18041640

**Published:** 2021-02-09

**Authors:** Lucía Morales-Sánchez, Violeta Luque-Ribelles, Paloma Gil-Olarte, Paula Ruiz-González, Rocío Guil

**Affiliations:** 1Department of Psychology, University of Cádiz, 11519 Cádiz, Spain; violeta.luque@uca.es (V.L.-R.); paloma.gilolarte@uca.es (P.G.-O.); paula.ruiz@uca.es (P.R.-G.); 2Institute for Research and Innovation in Biomedical Sciences of Cádiz (INIBICA), 11009 Cadiz, Spain; 3University Research Institute for Sustainable Social Development (INDESS), University of Cádiz, 11406 Cádiz, Spain

**Keywords:** breast cancer, women, body image, self-esteem, intervention, program, therapy, systematic review

## Abstract

Breast Cancer (BC) is the most common neoplasm in women worldwide, considered a global public health problem. Among BC women, some of the most common psychological symptoms in the adaptation to the disease are reduction in self-esteem and distorted body image (BI). Although there are numerous studies with the goal of promoting different psychological variables, BI and self-esteem are often separately observed despite their relationship and their importance in the process of the illness. Moreover, there have been no reviews that have synthesized the findings related to interventions aimed at enhancing both self-esteem and BI in BC women. Therefore, the objective of this review was to identify and examine the implemented interventions aimed at boosting both variables in this population. For this purpose, a systematic review was implemented following the PRISMA statement. A thorough search was performed on the following databases: Web of Science, PubMed, PsychInfo, PsychArticles, and Scopus. Among 287 records, only eight articles met the eligibility criteria. Interventions were grouped into three types according to their characteristics: Group therapies, Physical activity therapies, and Cosmetic and beauty treatments. The levels of effectiveness of the different interventions varied between them, and within each, in their impact on self-esteem and BI. More interventions focused on developing BI and self-esteem in this population are needed due to their ability to predict psychological functioning and quality of life of women with breast cancer.

## 1. Introduction

Breast cancer (BC) is one of the most common neoplasms in women, accounting for 16% of all female cancers and with over 1.2 million cases diagnosed each year worldwide [1], considered a global public health problem [2,3]. Due to improvements in diagnosis and treatment of the disease, BC has a survival rate of 90% in 5 years, and about 80% in 10 years, although survivors have faced multiple mental and physical health challenges [4,5,6]. Therefore, health professionals are concerned about the quality of life of female survivors, including physical, emotional, psychological, and social aspects related to trauma and adaptation to BC [7,8,9].

It has been observed that systemic treatments in BC women such as chemotherapy, hormonal treatment, and radiotherapy harm the quality of life of the patients [10,11]. These adverse effects are physical, (e.g., pain, vomiting, and sleep disorders) and psychological (poorly perceived body-image, depression, anxiety, etc.). These problems may persist for a long-term period after treatment completion [10]. For example, when BC women start chemotherapy, they face aspects such as hair loss, eyelashes, weight loss, etc. [11,12,13]. These physical losses and changes produce cognitive, behavioral, and emotional alterations, which affect the well-being of women, and their self-esteem [14,15,16]. This is because, for many women, self-esteem is based exclusively on the perception of their own body, so that poor perception of this can lead to a decrease in self-esteem and, at the same time, negatively affect a person’s daily life [17,18,19].

Specifically, among BC Women, the most common psychological symptoms in the adaptation to the disease include disturbance of mood, increased level of distress, distorted body image, and diminished self-esteem [9,20,21,22]. In this regard, body image (BI) refers to the perception, evaluation, and derived feelings about body appearance and physical functioning [23], considering it as part of self-concept [24,25,26,27]. BI seems to be a predictor of psychological functioning in BC women, i.e., high satisfaction with appearance and better BI levels predict better quality of life [28,29]. Self-esteem is linked to self-concept, and it concerns the attitudes or feelings of satisfaction with oneself, based on the evaluation of one’s characteristics [19,30].

Scientific literature highlights the relationship between BI and BC: its chronic nature, its epidemiological relevance, and the substantial psychological and social connotations it has for women because of the importance of breasts for them [19,31,32]. Part of this importance lies in the association between women’s breasts and the idea of femininity dictated by social and cultural systems, which stress the ideal of beauty for women (i.e., health, youth, and symmetry) [33]. For most women, breasts are one of the elements that define them as such, and the loss of these would mean the forfeiture of their femininity [34,35,36,37]. Moreover, the women’s breast is related to the sphere of sexuality, physical attractiveness, motherhood, and breastfeeding. Therefore, for many women who suffer from this disease, this fact may mean that they give up their desire to be a mother [38,39].

It has also been scientifically proven that the type of surgical intervention used constitutes a relevant factor for the BI of women affected by this disease. When the treatment implies a surgical process, the situation becomes harder, since the woman must face a significant loss of her body [39,40,41,42]. In this regard, BC women who have had a mastectomy can show emotional instability, diminished physical attractiveness perception, reduction in self-esteem, and partner relationship disturbances [11,43]. A greater dissatisfaction was also found with BI in women preparing for late reconstruction than those with immediate reconstruction [44]. Both patients who had a mastectomy, and patients who had breast reconstruction or implants, reported lower satisfaction with their breasts, BI, and sexual functioning than those who had undergone breast conservation therapy or autologous reconstruction of the breasts [45].

In addition to these impacts, and the consequence of them, these women must also start a completely different life adapting to other routines and activities. Therefore, it is necessary to promote intended actions to achieve an adequate adaptation to the changes that they will experience [5]. In this respect, it is essential to work on the self-esteem and BI of patients due to the negative impact of BC on BI and female self-esteem, as well as due to its importance in the process of the illness [19,46,47,48,49,50].

In this regard, previous studies have guaranteed the effectiveness of psychological group therapies and interventions in improving self-esteem and BI [51,52,53,54], as well as in other psychological and biological dimensions [55]. Group interventions are a powerful therapeutic tool that promote personal interactions, a significant element of psychological development [56,57]. Its effectiveness is in offering emotional support and motivation and reducing anxiety and depression by offering the opportunity to learn how other people have successfully managed the problems generated by cancer [58]. However, there are numerous interventions aimed at promoting BI and self-esteem independently, but not together.

On the one hand, the latest investigations aiming to boost BI in BC women are divided into (I) Group therapies: group therapy interventions [59,60], and a text-based online group therapy intervention [61]; (II) Physical activity therapies: a medical dance/movement therapy for BI [62]; and (III) Cosmetic and beauty treatments: cosmetic programs [63], and beauty treatments [64].

On the other hand, the latest studies focusing on the improvement of self-esteem with other outcome variables are: (I) Group therapies: a Self-esteem/Social Skills and Cognitive-Behavioral Therapy [65], and a randomized education trial [66]; (II) Physical activity therapies: Physical Activity Intervention [67], Exercise Program interventions [68], and a Self-managed home-based moderate-intensity walking intervention [69]; and (III) Spiritual Interventions: a Mindfulness-Based Program [70], a Qigong Mind-Body Exercise [71], and a RIME intervention (Relaxation, Mental Images, Spirituality) [72].

Thus, several studies have intervened on BI or self-esteem independently, as well as with other outcome variables. Notwithstanding, from our data knowledge, there have been no reviews that have synthesized findings concerning interventions aimed to enhance both self-esteem and BI in BC women; hence the relevance of this scoping review to unify and connect the literature on interventions intended to promote these aspects in BC women. In this way, this study will provide relevant information to the scientific community, therapists, and health care professionals about how BI and self-esteem of BC women play a fundamental role throughout the disease, and after, as well as the need to intervene in both to improve the well-being and quality of life of these women. Hence, the objective of this review is to identify and examine the implemented interventions aimed at boosting both self-esteem and BI in BC women.

## 2. Materials and Methods

### 2.1. Data Sources and Search Strategy

For the implementation of this systematic review the PRISMA (Preferred Reporting Items for Systematic Reviews and Meta-Analysis Guidelines) statement was followed, which provides a stringent process to carry out such scientific works as systematic reviews. It consists of a 27-item checklist and a four-phase Flow Diagram (see Figure 1), with the goal of helping authors improve the reporting of systematic reviews and meta-analyses [73].

First, a thorough search was performed on the following databases: Web of Science (WOS), Scopus, PubMed, PsycINFO, and PsychArticles, using Key Words and Boolean operators with the ensuing combination: (“Breast Cancer” OR “Breast Cancer Women”) AND “Self-esteem” AND “Body image” AND (“Therapy” OR “Intervention”). All the databases were searched from inception to 8th October 2020. There were no restrictions on date but there were on language. In all data sets, filters concerning “English” and “Spanish” languages were applied. Titles, abstracts, and full manuscripts were peer-reviewed for inclusion in the present review.

Table 1 shows databases used, search strategy use, and the results obtained each.

### 2.2. Eligibility Criteria

Information concerning participants, interventions, comparison, outcomes, and study design (PICOS) is showed in Table 2.

The articles included in the review had to be full-text research articles published in English or Spanish that met the following inclusionary criteria: (1) the population was composed of women with BC; (2) the outcome variables included both self-esteem and BI; and (3) the articles comprised an intervention aimed to enhance women’s self-esteem and BI. Exclusion criteria involved: (1) non-English-or-Spanish-language articles; (2) participants of the study were not BC women (i.e., the population had other types of cancer or other diseases); (3) in the study BI and self-esteem were not used as dependent variables, but others were (e.g., sexuality, quality of life, anxiety); (4) there was no intervention identified in the study; (5) systematic, literature, annual and clinical review; books; unpublished articles; doctoral dissertations; commentaries; meeting and conference abstracts and proceedings.

### 2.3. Data Collection Process

After introducing the aforementioned Key Words and Boolean operators, as well as the filters concerning language in all Databases, a total of 287 articles were found. Finally, eight studies fulfilled the eligibility criteria and were included in the Systematic Review.

Figure 1 shows the overview of the selection process through the Flow Diagram, in which data was pulled out systematically.

### 2.4. Analysis

To summarize the research on the interventions focused on the improvement of the BI and self-esteem among women with BC, a qualitative synthesis was performed.

## 3. Results

### 3.1. Findings

Following the search strategy, a total of 287 articles were identified. However, 71 records were duplicated. Therefore, from a total of 216 articles, 179 were excluded following the reading of titles and abstracts. Of all of these, a total of 37 were selected following eligibility criteria. Among these, 29 works were deleted and finally a total of 8 studies were selected to be included in the qualitative synthesis. The list of the excluded articles (*n* = 29) and the reasons for exclusion are provided in Appendix A.

### 3.2. Study Characteristics

The general characteristics of the studies listed in the review (n = 8) can be found in Table 3. Articles were published from 1999 to 2019 in a variety of scientific journals with different aims and scopes: Archives of General Psychiatry [74], Body Image [75], Psicooncología [76], European Journal of Cancer Care [77], Psycho-Oncology [78], The Arts in Psychotherapy [9], Clinical and Health [19], and Revista Brasileira de Medicina do Esporte [35]. Studies involved participants from Brazil [35], the United States of America [9,74], the United Kingdom [75], Spain [19,76], South Korea [77], and Austria [78]. All the records were written in English except one that was in Spanish but had its abstract in English [76].

#### 3.2.1. Design of the Studies

Half of the studies were Randomized Controlled Trials (RCT), two of them with pre-/post-design [9,76], and two with pre-/post- and follow-up design [74,78]. Three of the studies were Non-Randomized Controlled Trials with pre-/post- design [35], and with pre-/post- and follow-up design [19,77]. Only one of the studies did not have a control group, but the sample was randomized [75]. Therefore, it presented a quasi-experimental pretest-posttest and follow-up design.

#### 3.2.2. Participants and Regrouping

A total of 502 BC women were included in the investigations. The number of participants in the different studies ranges from 19 to 312 women with BC. Study 1 [35] included 19 women and divided them into a Control Group (CG) (*n* = 8) and an Intervention Group (IG) (*n* = 11). Study 2 [9] was composed of 33 women who were separated because of their precedence (North and South) and were allocated into two IGs, both composed both of 10 to 12 women, and two CGs. Study 3 [74] contained 312 women divided into a total of 28 groups: 7 CG (*n* = 77), 7 Education groups (*n* = 79); 7 Peer discussion groups (*n* = 74); and 7 combination groups (*n* = 82). Study 4 [75] was composed of 22 women divided into two IGs who received the intervention. Study 5 [76] included 38 women with a BC diagnosis divided into a CG (*n* = 19) and an IG (*n* = 19). Study 6 [77] was performed with 60 patients who composed an IG (*n* = 31) and a CG (*n* = 29). Study 7 [78] had 44 BC patients allocated to either IG (*n* = 22) or CG (*n* = 22). Finally, Study 8 [19] included a total of 188 women divided into a CG (*n* = 81) and an IG (*n* = 107).

#### 3.2.3. Interventions and Professionals Who Implement These

The interventions included in the different studies can be classified into different modalities according to their characteristics: 

Group therapies: An Education and Peer Discussion group intervention [74] implemented by an oncology nurse and an oncology social worker; a Psychosocial intervention program [19] executed by two female psychologists; and two Cognitive-Behavioral therapy interventions [75,76] were detected. Of these last two, the first was carried out by a clinical psychologist and peer BC nurse specialists, but the second did not mention this aspect in the study.

Physical activity therapy: A Belly dance intervention [35] and a Dance/Movement Therapy [9] were carried out with BC women. In the first facilitators were not mentioned, but in the second the intervention was carried out by dance/movement therapists who teach Authentic Movement locally, nationally, and internationally.

Cosmetic and beauty treatments: a Cosmetic education program [77] and a Beauty Care intervention [78] were found. Both were carried out by professional beauty specialists.

#### 3.2.4. Changes in BI and Self-Esteem after the Interventions

Some of the studies found the interventions effective in terms of significant improvements in both self-esteem and BI variables after implementation [19,74,75,78]. Other studies, however, only found changes in one of the variables, i.e., interventions produced positive changes in BI but not in self-esteem [35], or improved self-esteem but not BI [76]. Otherwise, some of the studies did not obtain significant results after the interventions, therefore they were not effective [9,77].

In Study 1 [35], significant differences were observed between pre- and post-intervention in the BI variable, but not in self-esteem. Spontaneous reports during the classes confirmed an improvement of self-perception of femininity and confidence in women. Study 2 [9] did not obtain significant results concerning BI and self-esteem, but did in other variables. Therefore, the intervention had no effect. However, qualitative analyses revealed that, after the Authentic Movement Therapy Group, participants showed strong self-perceived improvement in BI and self-esteem, i.e., an increase in awareness, acceptance, and in appreciation of the body and the self.

Results in Study 3 [74] showed that, before group interventions, patients in Education groups showed higher self-esteem and better BI. However, there was no evidence of benefits from Peer discussion group interventions. In Study 4 [75], significant improvements were identified after the intervention, or emerged at one-month follow-up, on the majority of the BI measures, except for body-related self-care attitude, and in self-esteem. From qualitative results it was extracted that most participants felt the intervention had improved their BI and it was beneficial for them.

Study 5 [76] revealed that, after treatment, statistically significant beneficial differences were found between the intervention group and the control group in self-esteem. However, the improvements observed in BI were not statistically significant. Moreover, Study 6 [77] found that the intervention had no significant effects on the self-esteem and BI of participants.

Study 7 [78] showed a positive effect on BI and self-esteem of women after the intervention. Patients in the intervention group reported higher self-esteem than before the intervention, as did the control group. Moreover, this was maintained in the follow-up. However, both groups reported increased BI, irrespective of intervention. These changes decreased as the follow-up passed. Finally, Study 8 [19] revealed that participants in the psychosocial program showed more positive BI and higher self-esteem than women who did not participate.

#### 3.2.5. Methodology and Measurements

All the studies used quantitative measures. However, only two used a qualitative methodology, combining both quantitative and qualitative methods [9,75]. Notwithstanding, it might be highlighted that qualitative data is reported quantitatively and analyzed in an unclear way in some cases. It seems to serve as simple support of the extra extracted quantitative data. In Study 4 [75], the qualitative feedback was collected at post-test only. Therefore, these data could not be triangulated.

Concerning instruments used, in Table 3, those regarding BI and self-esteem have been selected, but not those that assess other outcome variables in the same study. In this respect, some studies agreed on the instruments to assess self-esteem. The majority used the Rosenberg Self-esteem Scale (RSE) [79]. However, high variability in the employment of instruments to assess the BI variable was found, although it has been observed that the most commonly used is the Body Image Hopwood Scale (BIS) [80].

Regarding self-esteem, six of eight studies assessed this outcome variable with the Rosenberg Self-esteem Scale (RSE) in its original version [79] in the majority [19,74,76,77], as well as in adapted versions of this instrument [81,82] in two studies [35,78]. Other studies used the Single-Item Self-Esteem Scale [83] to assess global self-esteem [75] and the Berscheid, Walster, Bohrnstedt Body-Image Scale (BWB) [84] to assess both BI and self-esteem [9].

Concerning BI measures, there is no such consensus. Four studies [19,75,76,78] used the Body Image Hopwood Scale (BIS) [80], which was created for use with cancer patients. One of these studies [75], apart from the use of BIS, assessed BI with such other instruments as the Appearance Evaluation Subscale (MBSRQ) [85]; the Weigh and Shape Concern Subscales (EDE-Q) [86]; the Social Activities and Clothing Subscales (BIAQ) [87]; the Body Appreciation Scale [88]; the Cognitive Reappraisal Subscale (PARCA) [89]; and the Body-Related Self-Care Scale: Attitude Scale [89]. Besides, one of the studies [35] evaluates BI with the Body Image Questionnaire after Breast Cancer (BIBC) in the Brazilian adaptation [90]. The use of the Body Cathexis Scale (BCS) [91] in the cosmetic program education was also observed [77], and an adapted version of the Cancer Rehabilitation Evaluation Systems (CARES) [92] in the Education and Peer Discussion group interventions [74]. Finally, Dibbell-Hope [9] utilized in her Dance/Movement Therapy study the Berscheid, Walster, Bohrnstedt Body-Image Scale (BWB) [84] to assess both BI and self-esteem, as aforementioned.

#### 3.2.6. Characteristics of the Participants

To examine the different studies, the sociocultural and personal characteristics of participants should be a relevant element to take into account. The women of Study 1 [35] had to be aged between 40 and 80 years (age mean: 54.55 years). Exclusion criteria were also determined by educational level, by classification as being illiterate, and including those who presented with stage IV BC [93]. It must be highlighted that most women of the IG were housewives, while most women of the CG were retired, unemployed, or on medical leave. Otherwise, Study 2 [9] included a higher range of women’s age (35 to 80; mean: 54.7 years) and selected women came from two different geographical and demographical zones of the San Francisco Bay Area. Namely, the total sample was represented by women from an urban, sophisticated, and heterogeneous area (Northern), as well as by women from a suburban, conservative, and homogeneous area (Southern). More than 75% of them had Stage I cancer at diagnosis and 81% had had a modified radical mastectomy as the primary treatment while 21% had received chemotherapy, 19% radiation, and 10% reconstruction.

In Study 3 [74], the age of the participants ranged from 27 to 75 years (mean: 48.25 years), and a high percentage (69%) had stage II disease. Moreover, most women underwent lumpectomies (68%) rather than mastectomies (32%). From these 312 women, the majority were married (67%) or divorced (12%), white (93%), Catholic (49%) or Protestant (44%), and with high education levels: high school graduates (30%), some college (28%) and College graduates (76%). The characteristics of the women in Study 4 [75] included an average age of 51.55 years, and the majority were white (95.5%), married or in a relationship (72.7%), and all were parents. Most of them were employed (63.3%) and educated beyond secondary school (72.7%). The most common diagnosis in these women was Stage II (27.3%) or III (36.4%) cancer, and 59.1% had undergone mastectomy with reconstruction. Besides, most of them had received chemotherapy (68.2%), radiotherapy (77.3%), and hormonal therapy (86.4%).

Study 5 [76] added a large list of inclusionary criteria for participants’ selection. Women had to be in a disease-free situation, to have had a mastectomy in the last three years and to be, at the moment of the intervention, only on hormonotherapy. Women had to be between 30 and 60 years old. The total sample was composed of women with an average age of 50.2 years (range between 38 and 55 years). Most were married (73%), some were widows (13%) and 13% were single. In Study 6 [77], 31 women in the IG (age mean: 43.97 years) and 29 women in the CG (age mean: 45.41 years) who had undergone a surgical mastectomy in two years and were subsequently being treated with chemotherapy or radiation therapy participated. The most common characteristics of both IG and CG were married (83.9%; 89.7%), with children (74.2%; 93.1%), with university studies (67.8%; 55.1%), and the majority had received chemotherapy (74.2%; 82.2%) followed by radiotherapy (32.3%; 37.9%), respectively.

The general characteristics of women who participated in Study 7 [78] included having had a diagnosis of early BC, 18 years old and above, reporting appearance-related side effects of cancer treatment (e.g., irritated or pale skin, loss of scalp hair, eyebrows, or eyelashes), and time since diagnosis less than 24 months. The majority of women in the IG (age mean: 39.6 years) and in the CG (age mean: 37.4 years) were married (45%; 36.8%), followed to be in a relationship (20%; 36.8%) and single (20%; 15.8%), respectively. Just over half of the women (60%; 52.6%) had children and the majority were on sick leave (80%; 78.9%). Most had received chemotherapy (90%; 84.2%) followed by radiation therapy (20%; 31.6%), and 25%–42.1% had a mastectomy.

Finally, participants in Study 8 [19], were aged 27 to 65 years (age mean: 48 years). The majority of women were married (78.2%), were educated to elementary or primary level (45.2%), and most of them were employed (52.6%). Of the total sample, 69% had had a tumorectomy and 39.4% underwent a mastectomy.

## 4. Discussion

Among BC women, the most common psychological symptoms in the adaptation to the disease include distorted BI, and diminished self-esteem [9,20,21,22]. In this respect, it is essential to work on these two aspects due to the negative impact of BC on BI and female self-esteem, as well as due to its importance in the process of the illness [19,46,47,48,49,50].

This is possibly the first study that has reviewed the scientific literature on interventions to promote both self-esteem and BI in BC women. Therefore, the objective of this systematic review was to identify and examine these and map out the work done in the field. We found only few studies (n = 8 articles) that matched with the purpose of the study, and it was observed that the implemented interventions were diverse. All were group interventions, but we classified them into group therapy interventions (n = 6), physical activity therapies (n = 2), and cosmetic and beauty treatments (n = 2) based on their characteristics. All of the studies showed a quantitative methodology, but two complemented this with qualitative methods. Moreover, it was noted that most of the studies (seven out of the eight) had a control group and intervention/s group/s. Half of the studies (four out of the eight) were randomized controlled trials (RCT), three were non-randomized controlled trials (NRCT) and one had a quasi-experimental pretest-posttest and follow up design, without a control group but with a randomized sample. Characteristics of the samples and the results of the interventions were also examined. Regarding the last, although changes in both self-esteem and BI variables were obtained in all the studies after the interventions, not all of them found these changes significant. It was also found that the most commonly used instrument to assess BI was the BIS [80], and to evaluate self-esteem the RSE [79].

The levels of effectiveness of the different interventions varied between, and within each, in their impact on self-esteem and BI. Only half of the studies [19,74,75,78] showed significant improvements in both variables after interventions. Notwithstanding, one displayed that the improvements in the BI variable after intervention were independent of the study [78], and in another one of the types of intervention (peer discussion groups) did not have a significant impact [74]. However, the other half revealed that only one, BI or self-esteem, significantly changed [35,76], or that the improvements in both variables were not significant, so the intervention was not effective [9,77]. These results could be influenced by sample size, the professionals applying the interventions, the methodology and instruments selected for the study, the extent and duration of the intervention, and the covered content. The studies used self-reported measures, which could be a limitation in the obtained results. Therefore, analysis of the self-reported measurements for accuracy is suggested. For instance, in Villarini et al. [94] it was demonstrated that self-reported measures and measurements performed by professionals in breast cancer women showed no differences between the two above mentioned methods. This stands in contrast to Study 2 [9], in which the two methods obtained different results. It is also recommended that future research carry out randomized controlled trials to eliminate possible bias in the effect of the interventions and to avoid limitations [35,75].

As mentioned above, the interventions were classified into different modalities according to their characteristics. The first refers to Group Therapies, and it includes a Psychosocial Intervention Program [19], an Education and Peer Discussion Group Intervention [74], and two Cognitive-Behavioral Therapy Interventions [75,76]. In general, these four interventions obtained positive results. This finding is congruent with the results obtained in Brandão & Mena’s systematic review [55], which after analyzing numerous psychological intervention programs for women with BC, concluded that the majority of the interventions (cognitive-behavior therapy, supportive-expressive therapy, psychoeducation, and psychosocial therapy) had positive results in psychological and biological dimensions, and only a minority of these interventions did not report any benefits. In this respect, studies focused on the effectiveness of group interventions in health issues have shown better results when they address issues experienced by the participants [95]. Sharing experiences and awareness in small groups is also essential for women, because they learn from each other, leave their private spheres, politicise their daily experiences, and identify themselves in terms of gender [96].

Concerning the content of the interventions, some studies address sociocultural elements, the ideal of beauty and femininity, relations and intimacy, and people’s reactions, incorporating, therefore, social and structural elements at work on BI [75]. However, in other studies, when BI and self-esteem are addressed, the sociocultural determinants were not mentioned, e.g., in the module focused on sexuality [76]. It is suggested that future interventions explicitly report from a feminist approach on the sociocultural influences on BI and self-esteem. Stories should be provided from women with diverse experiences concerning both variables, addressing the effects of treatments such as hormone therapy on sexual desire. Similarly, it would be of interest if future interventions informed on the influence of environmental and occupational factors on the disease [97], or of the market created around this [98]. Thus, other visions of BC and other coping strategies that would foster psycho-political development could be offered [99], along with a social feminist identity that could face gender mandates and their perpetuation [33,100,101]. The results of these studies are diverse in terms of the improvement of BI and self-esteem. Firstly, it the need to increase the sample size is indicated. The professionals that apply the interventions should be clinical psychologists. Besides, it should be examined if the improvement after group interventions derives from the content of the sessions or from the group work itself. Finally, convenience in strengthening the already existing support networks rather than creating new ones should be studied.

The second aggrupation refers to Physical activity therapies. A belly dance intervention [35] and a Dance/Movement Therapy [9] were carried out with BC women. These interventions showed diverse results concerning their impact on BI and the self-esteem of these women. Likewise, in the same study, different results were obtained according to whether a qualitative or quantitative methodology was applied. While Study 1 [35] obtained positive results in BI, but not in self-esteem, those obtained in Study 2 [9] did not confirm the efficacy of Dance/Movement Therapy on these variables. This agrees with results obtained by Bradt et al. [102], which did not find evidence for an effect of dance/movement therapy on body image in cancer patients. For future interventions, the need to control sample loss and to increase the size and diversity of the sample is indicated. It is also suggested that the facilitators or professionals should be the same people to control possible variations due to leadership styles. Likewise, in the case of qualitative methodology, it is recommended to control for social desirability in the answers. Likewise, methodological limitations and weaknesses in the interpretation of the results have been detected.

Cosmetic and beauty treatments are the third group of interventions that compose the present study. This group includes such interventions as a cosmetic education program [77] and a beauty care intervention [78]. Although these interventions are partially effective, they can perpetuate the idea that there is only one form of femininity, linked to the ideal of young, healthy, and symmetrical bodies, that must be fulfilled, and that women must be feminine. Furthermore, they tend to unify and standardize women’s bodies, making their variability invisible [32,98,103]. Moreover, it must be remarked that both interventions’ duration was a one-day session. Therefore, the obtained results should be considered with caution. It is suggested that interventions along these lines incorporate information about gender mandates and the ideal of female beauty and its impact on women’s health, as well as to amply the extension of the program to obtain realistic outputs, sustained over time.

From the obtained results, it is extracted that group therapies and psychological interventions seem to have positive results on BI and self-esteem of BC women. Physical activity therapies and dance interventions, as well as cosmetic and beauty treatments, could also predict improvement in these variables, but it is suggested to combine these modalities with group therapies to verify if their combination shows greater results. Moreover, it is recommended to take into account all the previous recommendations regarding duration, extent, contents, methodology, and sample size.

Regarding the duration of the interventions, except for two [35,76], the majority of the studies (six out of eight) informed about the impact of the intervention after specific measurement points. Specifically, one [9] did not show positive effects in post-treatment or three weeks after the intervention. Likewise, another study [74] reported positive results six months after the intervention in mental and physical functioning of the women who attended the Education Groups, but not in those who participated in the Discussion Groups. In Study 4 [75], follow-up measures were performed after the implementation of the intervention. Better results in BI and self-esteem were found. At follow-up, only a significant improvement in distress was observed. In two of the studies [77,78], follow-up measures were performed at one month and at eight weeks, respectively, after the interventions. In the firs, improvements over time in BI and self-esteem were not reported, and in the second, better self-esteem levels were reported at eight weeks post-treatment. In another study [19] measures were carried out at post-treatment and 6 months’ follow-up. BI measures increased progressively from the first measure to the 6-month follow-up. However, self-esteem decreased post-treatment but reported a new increase at follow-up.

It is observed that in two studies [19,74] follow-up measures were performed six months after the treatment, as the Society of Prevention Research Criteria for Efficacy recommend [104]. Moreover, in Helgeson et al. [105], distinct trajectories of change in psychological and physical adjustment to breast cancer were exhibited over four year in the study 3 sample [74]. Likewise, follow-up measurements were carried out at different points in time (e.g., at 1, 2, and 3 months) in studies 2, 4, 6, and 7 [9,75,77,78]. Any case, studies that include some measure of follow-up represent an advance, as this element is not always incorporated [75]. This leads us to point out the importance of conducting studies that explore the long-term impact of the intervention.

Concerning the instruments used, most of the studies (six out of eight) agree with the instruments that assess self-esteem, using the RSE in its original version [79] in the majority [19,74,76,77], as well as adapted versions of this instrument [81,82] in two studies [35,78]. The BWB- [84] and the Single-Item Self-Esteem Scale [83] were also used in two out [9,75] of the eight studies.

However, there is no such consensus and high variability in employed instruments regarding the BI variable. Although half of the studies [9,75,76,78] used the BIS [80], which was created specifically for BC patients, the others evaluated BI with other scales, i.e., the BIBCQ [90], the BCS [91], the CARES [92], and the BWB [84]. Moreover, one of the studies [75], apart from the BIS, also assessed BI with the MBSR [85], the EDE-Q [86], the BIAQ [87], the Body Appreciation Scale [88], the PARCA [89], and the Body-Related Self-Care Scale: Attitude Scale [89]. Therefore, this aspect can affect the generalization of the results and can lead to methodological and conceptual difficulties, because BI partly coincides with sexuality and also with the broader concept of self-image and self-concept [19,106,107]. It is also suggested that the examined population be taken into account. In this case, only two of the identified scales among the studies coincide with the six BI scales for cancer patients found in the prior review of Annunziata et al. [49], i.e., the BIS and the BIBCQ. It is recommended that future works select questionnaires and other instruments according to the study sample.

All the studies used quantitative measures. However, only two combined both quantitative and qualitative methods [9,75]. However, in one [9] the results were reported quantitatively (using percentages), and in the other quotations were presented without explanation of their meaning. In this regard, the study of Richard et al. [78], included in the present review, suggested that future studies should compile qualitative data on the subjective experiences of the participants, as well as mechanisms for improvement. It suggested also that future mixed studies incorporate strategies for the triangulation of information. Studies that adopt a feminist approach also point to the need to use qualitative methodologies to understand the multiple experiences of BC [32,103,108].

Regarding the characteristics of the participants, studies in general gave information about their civil status. However, except for two studies [75,78], information about whether the participants maintain affective-sexual relationships was invisible. Moreover, one study explicitly excluded illiterate women [35], while the others did not specify literacy. Besides, several works indicated the exclusion of women with hallucinations, delusions, other clinical symptoms indicating the advisability of inpatient hospitalization, and history of hospitalization for psychiatric illness, and another excluded those presenting previous psychopathology that requires specific medication for its treatment, or presence of active disease, and finally, another work excludes intellectual impairment, that would prevent completing the self-report measures, and history of a psychiatric disorder. Two of the studies included information regarding the racial groups to which the participants belong [74,75]. This coincides with works that allude to the idea that the dominant culture has made women’s experiences invisible and unified them by assimilating them into patriarchal models of reality [32,103]. From the perspective of feminism, the need to reclaim women’s experiences is urged [109], trying to make diversity visible concerning differences in race/ethnicity, class, and sexual identity [106]. Future work could also address women in vulnerable situations, such as illiterate and mentally ill women.

Five out of the included studies [74,75,76,77,78] did not give information about the influence of the general, demographic, and/or clinical characteristics of participants on the intervention effectiveness. Likewise, although one of the studies [35] did not take into account the participants’ profiles, it refers to a previous study [93] where the women’s general and clinical characteristics were reported. Moreover, other among the included studies [9] indicated that older women, women at earlier stages of cancer, with less invasive treatments, who had completed treatment long ago, and with more experience of physical activities tended to have a more positive mood and less distress after treatment than others, but not better BI. However, younger women, especially those with prior experience with sports and dance, tended to have an improved BI. It was also found that age seemed to have no relation with the type of medical treatment (mastectomy vs. lumpectomy) and body satisfaction. Other studies [19] reported that sociodemographic aspects such as age, marital status, educational level, and work status had no effect on BI or self-esteem at any of the measurement points. This finding contradicts the literature that affirms that age and marital status are associated with these variations [110,111]. Concerning the type of surgery, women with tumorectomy showed more positive BI than those with mastectomy, and this was repeated at all points of the measurements, as well as in the different groups. This finding is in line with the results of previous studies [11,43,44,45,112,113].

The strengths of this systematic review are that this is possibly the first work that maps out the evidence for interventions aimed to enhance both self-esteem and BI in BC women. For that purpose, a thorough search of several databases was carried out. Articles were examined and exhaustively screened according to the eligibility criteria. Moreover, two of the most commonly used languages worldwide were eligible and articles were exhaustively examined. However, this study is not exempt from limitations. We recognized that some studies were missed because they had not made full-text available and some were conference abstracts or proceedings, among other reasons for exclusion. Some of the studies showed a mild risk of bias in descriptions of the interventions or in the sample, or were not randomized trials. Besides, this review included only studies in which both variables, BI and self-esteem, were examined in the same study. In further research, it could be interesting to include studies that examine both variables separately in BC women. This could allow a mapping out of the characteristics and results obtained, to extend scientific knowledge about the topic, and classify the instruments most commonly used to assess BI and self-esteem in BC women. Moreover, the present study has not been previously registered in any review protocol database, which may offer additional scientific information about the work. A meta-analysis of these studies would also be appropriate, and more interventions that enhance BI and self-esteem in BC women are needed due to their importance in the process of the disease, as well as to their ability to predict the psychological functioning and quality of life of this population [28,29,55].

## 5. Conclusions

The objective of this review was to identify and examine the implemented interventions aimed to enhance both self-esteem and BI in BC women. Its principal contribution was to provide an overview of the interventions carried out to improve both variables in women with BC. Three types of interventions were grouped according to their characteristics: Group therapies (education and peer discussion groups, cognitive-behavioral therapy interventions, and a psychosocial intervention program), Physical activity therapies (mainly focused on dance and movement), and Cosmetic and beauty treatments. The levels of effectiveness of the different interventions varied, between each other, and within each, in their impact on self-esteem and BI. Its principal strength is that this is possibly the first work that maps out the evidence for interventions focused on improving both self-esteem and BI in BC women. Its main limitation is that some studies were missed because they had not made full-text available, or were conference papers. For future interventions, an increase in the diversity of the profiles of women that are incorporated, the standardization of measurement instruments, especially those related to the assessment of BI, the incorporation of qualitative evaluations, and the triangulation of these with quantitative measurements are suggested. It is also considered of interest that the interventions include contents that critically analyze social pressures to fulfill the ideal of femininity (e.g., healthy, young, and symmetrical bodies) and strategies to get rid of the need to fulfill these mandates. It is also proposed to investigate the relationship between internalized sexism, social feminist identity [101], and the results of interventions aimed at boosting self-esteem and BI. The combination of group psychological therapies with physical activity therapies and cosmetic and beauty treatments is recommended. More interventions focused on the development of positive BI and self-esteem in this population are needed due to their importance in the process of the disease, as well as their ability to predict the psychological functioning and quality of life of women with BC.

## Figures and Tables

**Figure 1 ijerph-18-01640-f001:**
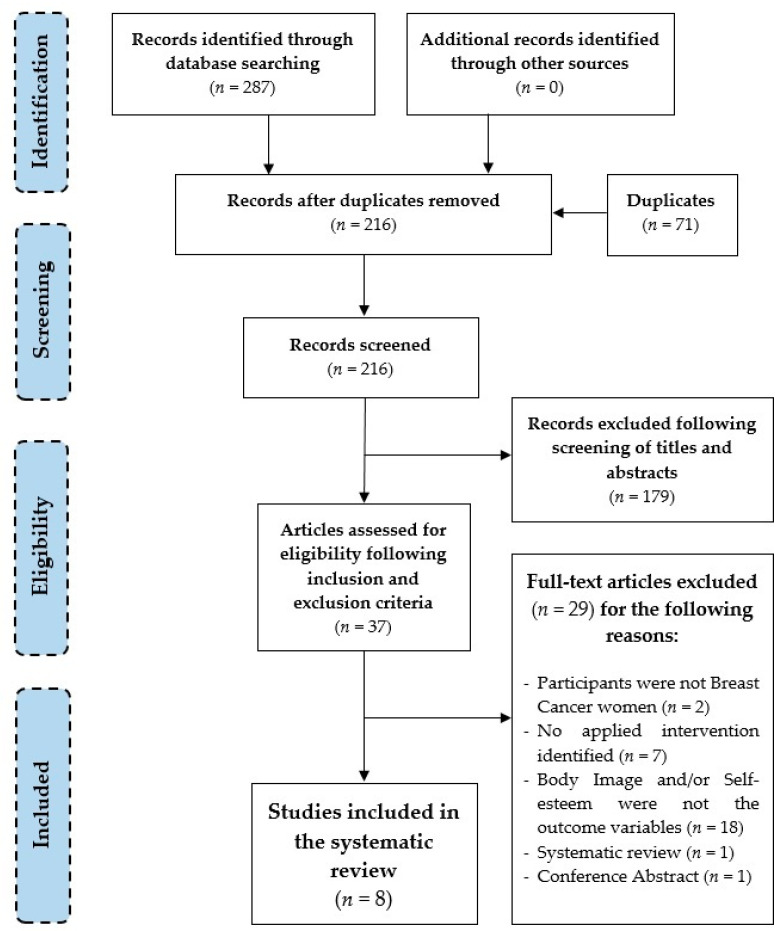
Flow diagram of the selection process of the studies.

**Table 1 ijerph-18-01640-t001:** Search strategies used.

Database	Search Strategies
Web of Science (WOS)	Topic: ((“Breast Cancer” OR “Breast Cancer Women”) AND “Self-esteem” AND “Body image” AND (“Therapy” OR “Intervention”)) Refined by: Languages: (ENGLISH OR SPANISH) Results: 68 records
Scopus	TITLE-ABS-KEY ((“Breast Cancer” OR “Breast Cancer Women”) AND “Self-esteem” AND “Body image” AND (“Therapy” OR “Intervention”)) AND (LIMIT-TO (LANGUAGE, “English”) OR LIMIT-TO (LANGUAGE, “Spanish”)) Results: 47 records
PubMed	(“Breast Cancer” OR “Breast Cancer Women”) AND “Self-esteem” AND “Body image” AND (“Therapy” OR “Intervention”)) Filters: English, Spanish Results: 30 records
PsycINFO	((“Breast Cancer” OR “Breast Cancer Women”) AND “Self-esteem” AND “Body image” AND (“Therapy” OR “Intervention”)) AND la.exact(“ENG” OR “SPA”) Results: 25 records
PsycArticles	((“Breast Cancer” OR “Breast Cancer Women”) AND “Self-esteem” AND “Body image” AND (“Therapy” OR “Intervention”)) AND la.exact(“ENG” OR “SPA”) Results: 117 records

Notes: ABS (Abstract); KEY (Key Words); la.exact (Language); ENG (English); SPA (Spanish).

**Table 2 ijerph-18-01640-t002:** PICOS: Participants, Interventions, Comparisons, Outcomes, and Study design.

	P	I	C	O	S
I	Women with BC (early diagnosed, in treatment or long-term patient, survivors)	Psychological, physical, spiritual, beauty care, arts and dance interventions or therapies	CG or comparison between pre-post after intervention in the IG	Improving self-esteem and BI	Experimental and quasi-experimental design with pre-post measures
E	Participants with other type of cancers, or other illnessess	Surgery interventions and systemic BC therapies	No results of comparisons	Other outcomes/Self-esteem or BI separately with other outcomes	Observational, comparative, and validation scale studies/Systematic reviews/Meta-analyses/Clinical guides/Study protocols

Notes: I (Include); E (Exclude); BC (breast cancer); CG (control group); IG (intervention group); BI (body image).

**Table 3 ijerph-18-01640-t003:** Characteristics of included studies.

Study	Design	Scope	Participants	Measures	Details of Implemented Interventions	Results
Study 1. Carminatti et al., (2019) [35]	NRCT Pretest and posttest design	Influence of belly dancing on BI and self-esteem.	19 women diagnosed with BC (m.a. = 54.55) 11 in IG (m.a.: not mentioned) 8 in CG (m.a.: not mentioned)	Body image: BIBCQ (Body Image After Breast Cancer Questionnaire). Self-esteem: RSE (Rosenberg Self-Esteem Scale).	Type: Belly dance intervention Classification: Physical activity therapy. Extent: 24 sessions in 12 weeks (2 sessions p/w). Duration: 1 h/session. Assess other outcome variables: no Contents: (1) Warm-up initial stretching; (2) Main part; (3) Relaxation. Who implements interventions: not mentioned	Significant differences were observed between pre- and post- intervention in BI, but not in self-esteem. Improvement of femininity and confidence were reported by spontaneous reports during the classes.
Study 2. Dibbel-Hope (2000) [9]	RCT Pretest and posttest design	Influence of AMTG in the levels of psychological adaptation; their sustainability over time.	33 women diagnosed with BC (mean age = 54.7) geographically separated according to where they lived into Northern (1) or Southern (2) areas. They were randomly assigned: 10–12 women in IG1 (m.a.: not mentioned) 10–12 women in IG2 (m.a.: not mentioned) CG1: not mentioned CG2: not mentioned	Quantitative: Body-image and Self-esteem: BWB (Borscht-Walter-Bohrnstedt Body-Image Scale). Qualitative: Interviews and Written program evaluation.	Type: Dance/Movement Therapy Classification: Physical activity therapy. Extent: 6 sessions in 6 weeks (1 session p/w). Duration: 3 h/session. Assess other outcome variables: yes. Contents: (1) Circle to check-in their life that week; (2) movement group; (3) discussion group and closing ritual. Who implements interventions: Therapists from the dance/movement therapy community who teach Authentic Movement locally, nationally and internationally.	Quantitative results: There were no observed significant improvements in BI and Self-esteem. CG1 reported greater dissatisfaction with BI than did the CG2 (sociodemographic differences) IG2 showed higher levels of BI (satisfaction with the body) than CG2 (effects pre-post intervention). Age and past experience with dance and sports predict satisfaction with BI and self-esteem. No relationship between type of medical treatment and BI. Qualitative results: After AMTG, participants showed strongly self-perceived improvement in BI and self-esteem (increase in awareness, acceptance and appreciation of the body and the self).
Study 3. Helgeson et al., (1999) [74]	RCT Pretest, posttest and follow-up	Effects of education-based and peer discussion-based group intervention in women with BC	312 BC women (m.a.: 48.25) randomly assigned: 79 women in Education Groups 74 women in Peer-Discussion Groups 82 women in Combination Groups 77 women in CG	Self-esteem: RSE (Rosenberg Self-Esteem Scale). Body Image: 14-item body-image scale based on the Cancer Rehabilitation Evaluation Systems (CARES).	Type: Education and peer-discussion group interventions Classification: Group therapy. Extent: 8 sessions in 8 weeks (1 session p/w). Duration: Education (45min./session); Peer discussion (60min./session); Combination (135min./session); CG did not assist to intervention groups. Assess other outcome variables: yes. Contents of sessions: Education (adverse effects of chemotherapy, nutrition, exercise, body image, relationships/sexuality, between others; Peer Discussion (promotion of feelings of acceptance and encouragement to expression of feelings and confrontation of problems; Combination (the combined intervention was a sequential combination, beginning with education and ending with peer discussion contents). Who implements interventions: an oncology nurse and an oncology social worker.	Before group interventions, patients of Education Groups showed higher self-esteem and better BI. There was no evidence of benefits from Peer discussion group interventions.
Study 4. Lewis-Smith et al., (2018) [75]	Quasi-experimental pretest-posttest and follow-up design	Effects of a CBT based intervention in BC survivors.	22 women (m.a.: 51.55 years) who had completed active treatment for BC. 9 in IG1 12 in IG2	Quantitative: Body dissatisfaction (cancer-specific): Hopwood Scale. Body dissatisfaction (general): MBSRQ Weight and Shape Concern: EDE-Q Body image avoidance: BIAQ Body appreciation: Body Appreciation Scale. Acceptance of aging-related appearance changes: PARCA. Body-related self-care-attitude: Body-Related Self-Care Scale. Appearance investment: Self-Objectification Questionnaire. Internalization of appearance ideals: SATAQ-3. Appearance comparisons: PACS- Body image-related avoidance of intimacy: RFH and CAR Intimacy Scale. Self-esteem: The Single-Item Self-Esteem Scale. Qualitative: Focus groups and interviews	Type: Cognitive-Behavioral Therapy based-intervention. Adapted BI intervention ‘Accepting your Body after Cancer’. Classification: Group therapy. Extent: 7 sessions in 7 weeks (1 session p/w). Duration: 2 h/session. Assess other outcome variables: yes Contents: Session 1 (body image), Session 2 (CBT, anxiety, body image and self-esteem); Session 3 (stopping negative body-related self-talk, balanced thoughts, self-care activity schedule, body function and movement; Session 4 (sociocultural pressures for women in midlife, internalization of the youthful-thin ideal, body comparisons, body nurture); Session 5 (exploration of relationships and intimacy; managing people’s reaction; cognitive restructuring process; physical activity and movement); Session 6 (Core beliefs, Modifying mistaken beliefs), Mindful eating, relaxation exercise); Session 7 (positive body affirmations, reducing the chances of a setback, dealing with a setback, future plans). Who applies interventions: Clinical psychologist and peer BC nurse specialists. Type: focus groups Number of groups: 6 Participants: 3–5 women each Duration: 3 h. including short breaks. Contents: opinions in relation to each session’s content. Who implements interventions: Clinical psychologist and peer BC nurse specialists.	Quantitative results: Significant improvements were identified at post-test and sustained, or emerged at 1-month follow-up, iin the majority of the BI measures and self-esteem. There were no significant improvements in body-related self-care attitude at either post-test or follow-up. Qualitative results: Most participants felt the intervention was beneficial and had improved their BI.
Study 5. Narváez et al., (2008) [76]	RCT Pretest and posttest design	Effectiveness of a CBT in BC Survivors.	38 women (m.a.: 50.2 years) who have been diagnosed with BC and have been operated on for a mastectomy in the last three years. 19 in IG (m.a.: not mentioned) 19 in a CG (m.a.: not mentioned)	Body image: Hopwood scale. Self-esteem: RSE (Rosenberg Self-Esteem Scale).	Type: Cognitive-Behavioral Therapy Classification: Group Therapy. Extent: 9 sessions in 9 weeks (1 session p/w). Duration: 1.5 h/session. Assess other outcome variables: yes. Contents: mood status (2 sessions), BI and self-esteem (4 sessions), sexuality (2 sessions) and closing with the tests (1 session). Who implements interventions: not mentioned.	After the treatment, there were observed statistically significant beneficial differences between IG and CG in self-esteem. However, while there were observed improvements in BI, they were not statistically significant.
Study 6. Park et al., (2015) [77]	Prospective NRCT Pretest, posttest and follow-up	Effects of a cosmetic program on patients who had undergone a mastectomy for BC.	31 women in IG (m.a.: 43.97 years) 29 women in CG (m.a.: 45.41 years)	Body image: BCS (Body Cathexis Scale). Self-esteem: RSE (Rosenberg Self-Esteem Scale).	Type: Cosmetic education program ‘Make up your life’ Classification: Cosmetic and beauty treatments Extent: 1-day session. Duration: 2 h. Assess other outcome variables: yes. Contents: useful skills, including skin care, facial massage, applying make-up, hair care for depilation, and dressing strategies to preserve physical appearance). Each woman received a free make-up kit and practiced each of the techniques. After applying their make-up, the patients and professionals discussed and shared experiences related to changes in their appearance through the program. Finally, participants took pictures of their made-over appearance. Who implements interventions: two professional beauty specialists.	There were observed positive changes in BI and self-esteem, but were not significant.
Study 7. Richard et al., (2019) [78]	RCT Pretest, posttest and follow-up	Effects of a beauty care intervention in patients with early BC.	20 women with BC in IG (m.a.: 39.6 years) 19 women with BC in CG (m.a.: 37.4 years)	Body image: BIS (Body Image Scale). Self-esteem: RSE (Rosenberg Self-Esteem Scale).	Type: Beauty care intervention. Classification: Cosmetic and beauty treatments Extent: 1-day session. Duration: 4 h. Assess other outcome variables: yes. Contents: makeup workshop, a photo shooting, and of receiving professionally edited portrait and upper-body photos Who implements interventions: professional beauty specialists.	The intervention had a positive effect on the women’s BI, and self-esteem. No differences between CG and IG were found at the pretest. At posttest 1 and 2, patients of IG reported higher self-esteem than at pretest, as well as than CG. This change was maintained in the follow-up measures. From pretest to posttests (1 and 2) both groups reported increases in BI, irrespective of intervention. These changes decreased as the follow-up tests were passed.
Study 8. Sebastián et al., (2008) [19]	NRCT Pretest, posttest and follow-up	Effectiveness of an intervention program psychosocial	188 women (m.a.: 48 years) 107 women in IG 81 women in CG	Body image: Hopwood scale. Self-esteem: RSE (Rosenberg Self-Esteem Scale).	Type: Psychosocial intervention program. Classification: Group therapy. Extent: 14 sessions in 14 weeks (1session p/w). Duration: 2 h/session. Assess other outcome variables: no. Contents: are divided in five blocks: (1) Preparation for Chemotherapy (1 session); (2) Education for Health (5 sessions); (3) Body image (5 partial sessions); stress management and personal coping skills (5 partial sessions) and communication skills and setting of goals (3 partial sessions). Who implements interventions: two female psychologists.	Women who participated in the program, at 6-month follow up, showed more positive BI and higher self-esteem than women who did not participate.

Notes: RCT (Randomized Controlled Trial); NRCT (Non-randomized Controlled Trial); BC (Breast Cancer); BI (Body Image); IG (Intervention Group); CG (Control Group); m.a. (Mean Age); AMTG (Authentic Movement treatment group); CBT (Cognitive-Behavioral Therapy); p/w (Per week).

## Data Availability

The data presented in this study are available online at https://www.mdpi.com/1660-4601/18/4/1640/s1 in Appendix A. List of the excluded articles and reasons for exclusion.

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
