# Peer review of "Enhancing Self-Esteem and Body Image of Breast Cancer Women through Interventions: A Systematic Review"

_ijerph, 2021, doi:10.3390/ijerph18041640_

Round 1

Reviewer 1 Report

This is an interesting research topic and the present systematic review might be of great help of future studies willing to measure Body Image (BI) and self-esteem in Breast Cancer (BC) research.

I have several comments for the present manuscript:

The main objective of the review was to identify and analyze the implemented interventions aimed at boosting both self-esteem and Bi in BC. It is not clear what the authors design with the term analyze? Except descriptions in the present systematic review there are no analyses.

I have an important concern about the eligibility criteria. First why excluding studies that included other type of cancer? Often in studies including different type of cancers, BC is the most common cancer. Further the results of studies including other types of cancers may very well have separate results for the different included cancers (and BC in particular).

Why did the authors decide not to include BI and self-esteem presented in quality of life scales? BI is an independent dimension of the very largely studied EORTC BR23. By restricting your inclusion criteria’s, you may most probably have missed important studies. Several manuscripts were selected in the presented search strategy but then excluded by the very strict eligibility criteria which shows the number of manuscripts decreasing from 217 to only 8 have.

Page 6/23 “Notwithstanding, although results were not statistically significant, it was observed that experimental group 2 (Southern EG) showed higher levels of BI than control group 2 (Southern CG) after the intervention.”  

Then repeated in the same page last sentence of the fifth paragraph (no more line numbering) “More-over, Study 6 [81] found positive changes in BI and self-esteem before the intervention, but these changes were not significant.”

In relation with the two previous sentences: If the BI was not statistically significant then the intervention has no effect! And in the second sentence it is not clear to what changes you are referring. If the changes were not significant, they cannot be called changes! The present review needs to criticize methodological weakness of included studies. The authors should not reiterate the incorrect results interpretations of the different included studies.

In addition to the very detailed description of the characteristics of the participants in each separate study. I was expecting also to read (or discuss) if those characteristics (age, education, urban/rural, BMI ethnicity…) and BC treatment characteristics were controlled for in the different interventions. For instance, was the impact of the interventions the same in young and older women, in those who received chemotherapy or not, … Even if these aspects are not discussed in the included studies it is still of importance to develop the discussion in this direction.

It looks that the authors use both the term intervention and experimental. Are these terms interchangeable? What the authors intend by the term intervention needs to be defined.

Table 3 I suggest including the study publication year and the number to facilitate the reading.

There is no discussion about the duration over time of the interventions impacts. How long are the effect of different intervention on BI and self-esteem supposed to last? This important topic needs to be discussed even if the 8 included manuscripts missed to discuss it. 

Line 116 change IB into BI

In table 1 it is not clear why the authors have not used the same strategy of search in all databases? For instance, why not using the same starting date?

The introduction is too long and there are several parts that are out of topic, such as lines 101 to 108. In relation to the lines they should be presented in the entire manuscript. There is an error in the number of pages.

Author Response

Dear Reviewer 1,

Firstly, thank you for your kindly recognition and your useful comments. The responses are the following:

Point 1: The main objective of the review was to identify and analyze the implemented interventions aimed at boosting both self-esteem and Bi in BC. It is not clear what the authors design with the term analyze? Except descriptions in the present systematic review there are no analyses.

Response 1:

By the term “analyze” we meant the examination of the different studies that met eligibility criteria for the present systematic review. Notwithstanding, since this term can be confused, the term analyze will be replaced by “examine”. With the term "examine", we mean to study the characteristic of the studies, as well as to describe and explain them.

Thank you for your suggestion. Please, see the changes in lines 24, 148, 371, 444, and 719).

Point 2: I have an important concern about the eligibility criteria. First why excluding studies that included other type of cancer? Often in studies including different type of cancers, BC is the most common cancer. Further the results of studies including other types of cancers may very well have separate results for the different included cancers (and BC in particular).

Response 2:

Studies that included other type of cancer were excluded to avoid bias concerning the findings, as well as to obtain results with women with breast cancer, not other type of cancers.

For instance, the study DOI: https://doi.org/10.3389/fpsyg.2017.01633 was excluded because the study sample included breast, gynecologic, brain, lung, hematologic, and colorectal cancers. Even being breast cancer one of the most prevalent cancer diagnoses in the sample, the results obtained in the study were general and did not show separate results for the different cancers included, as it would be adequate. In that case, it would have been included in our study.

The sample of the other full-text excluded DOI: 10.1016/j.jadohealth.2017.02.021 were adolescents with breast asymmetry. Although this study was included after reading the title and abstract and it met some inclusion criteria, when full-text was examined it was removed because the sample was not composed by breast cancer patients but by adolescents with breast asymmetry.

Thank you for your comment. I hope we have reply appropriately the questions and concerns.

Point 3: Why did the authors decide not to include BI and self-esteem presented in quality of life scales? BI is an independent dimension of the very largely studied EORTC BR23. By restricting your inclusion criteria’s, you may most probably have missed important studies. Several manuscripts were selected in the presented search strategy but then excluded by the very strict eligibility criteria which shows the number of manuscripts decreasing from 217 to only 8 have.

Response 3:

The number of manuscripts decreases considerably (to 287 to 8) because there were several duplicated manuscripts, and because there were numerous studies that were not related to the topic and that their outcome variables were others (please, see the table with the excluded articles and reasons, in supplementary materials).

Although it is relevant to us, to assess both outcome variables with independent and specifics instruments, if the studies would assess the outcome with the EORTC BR23, for example, and would meet the other inclusion criteria (breast cancer patients, an intervention was implemented, etc.) we would include in our review.

For instance, Study 2 assessed both BI and self-esteem with the same instrument (BWB (the Borscht-Walter-Bohrnstedt Body-Image Scale). As well, Study 4 assessed self-esteem with a single-item scale, and BI with the Hopwood Scale.

However, we have not had the opportunity to examine articles on which both self-esteem and body image were the outcome variables, and the body image variable was assessed by the EORTC BR23. During the selection process, there were articles on which quality of life was examined (and, in fact, body image and self-esteem inside of this construct), but the topic of the article was out of our aim, participants were not breast cancer women, or did not meet the other inclusionary criteria.

Finally, thank you for your interesting question. I hope we have adequately clarified your query.

Point 4: Page 6/23 “Notwithstanding, although results were not statistically significant, it was observed that experimental group 2 (Southern EG) showed higher levels of BI than control group 2 (Southern CG) after the intervention.”  

Then repeated in the same page last sentence of the fifth paragraph (no more line numbering) “More-over, Study 6 [81] found positive changes in BI and self-esteem before the intervention, but these changes were not significant.”

In relation with the two previous sentences: If the BI was not statistically significant then the intervention has no effect! And in the second sentence it is not clear to what changes you are referring. If the changes were not significant, they cannot be called changes! The present review needs to criticize methodological weakness of included studies. The authors should not reiterate the incorrect results interpretations of the different included studies.

Response 4:

We agree with the reviewer’s comments. Although the two paragraphs described the changes in both variables in the different studies, the way of explaining them was not the most appropriated. We should have been taking into account to not reiterate the incorrect results interpretations of the included studies.

In relation to the comments and suggestions, the following sentences have been removed and replaced by others:

  • “Notwithstanding, although results were not statistically significant, it was observed that intervention group 2 (Southern IG) showed higher levels of BI than control group 2 (Southern CG) after the intervention. Also, at baseline (pre-intervention), CG1 (Northern CG) reported greater dissatisfaction with BI than the CG2 (Southern CG). It can be due to sociodemographic characteristics. Besides, age and experience with dance and sports seemed to predict satisfaction with BI and self-esteem. This study did not found a relationship between the type of medical treatment and BI.”

Replaced by:Moreover, Study 6 [77] found that the intervention had no significant effects in the self-esteem and BI of the participants.” (please, see lines 317-319).

  • “positive changes in BI and self-esteem before the intervention, but these changes were not significant.”

Replaced by:Study 2 [9] did not obtain significant results concerning BI and self-esteem, but it did in other variables. Therefore, the intervention had no effect. However, qualitative analyses revealed that after the Authentic Movement Therapy Group, participants showed strongly self-perceived improvement in BI and self-esteem, i.e., an increase in awareness, acceptance, and appreciation of the body and the self”. (Please, see lines 298-304).

Moreover, in the discussion section was also replaced “or that there were positive improvements in both variables, but they were not significant” by “or that the improvements in both variables were not significant, so the intervention was not effective.(please, see lines 473-474).

Likewise, it has been included in the Discussion section the following sentence: “Likewise, methodological limitations and weaknesses in the interpretation of the results have been detected.(please, see lines 549-551).

Point 5: In addition to the very detailed description of the characteristics of the participants in each separate study. I was expecting also to read (or discuss) if those characteristics (age, education, urban/rural, BMI ethnicity…) and BC treatment characteristics were controlled for in the different interventions. For instance, was the impact of the interventions the same in young and older women, in those who received chemotherapy or not, … Even if these aspects are not discussed in the included studies it is still of importance to develop the discussion in this direction.

Response 5:

Following the reviewer’s suggestion, it has been added the following paragraphs in the discussion section (please, see lines 668-691):

“Besides, five out of the included studies [74-78] did not inform about the influence of the general, demographic and/or clinical characteristics of the participants on the intervention effectiveness. Likewise, although one of the studies [35] did not inform about the impact of the intervention taking into account the participants' profiles, it refers to a previous study [93] where the general and clinical characteristics of the women were reported. Moreover, other of the included studies [9] indicated that women who were older, had earlier stages of cancer, less invasive treatments, completed treatment longer ago, and had more past experience with physical activities tended to have more positive mood and less distress after treatment than others, but not better BI. However, younger women, especially those with prior experience with sports and dance, tended to have an improved BI. It was also found that age seemed to have no relation with the type of medical treatment (mastectomy vs. lumpectomy) and body satisfaction. Besides, other of the studies [19] reported that sociodemographic aspects as age, marital status, educational level, and work status had no effect on BI or self-esteem at any of the measurement points. This finding is opposite of the literature that affirms that age and marital status use to be associated with these variables’ variations [110,111]. Concerning the type of surgery, women with tumorectomy showed more positive BI than those with mastectomy, and it was repeated at all points of the measurements, as well as in the different groups. This finding is in the line with the results of previous studies [11,43,44,45,112,113].”

Point 6: It looks that the authors use both the term intervention and experimental. Are these terms interchangeable? What the authors intend by the term intervention needs to be defined.

Response 6:

The word “experimental” has been removed and replaced by the word “intervention” in the majority of the cases (e.g. “experimental group” to “intervention group”; “EG” to “IG”). The word experimental only remains in the case of the description of the study's design.

Moreover, in TABLE 2, concerning the research question (PICOS), the authors provide information about the term intervention, in order to define it and to facilitate the reading and the understanding (Please, see Table 2, p.5).

Point 7. Table 3 I suggest including the study publication year and the number to facilitate the reading.

Response 7:

In order to facilitate the comprehension and the reading, the publication year and the number of the study were included in Table 3. Thank you for your useful suggestion.

Point 8. There is no discussion about the duration over time of the interventions impacts. How long are the effect of different intervention on BI and self-esteem supposed to last? This important topic needs to be discussed even if the 8 included manuscripts missed to discuss it. 

Response 8.

With regard to the suggestion, it has been added the following paragraphs in the discussion section (please, see lines 576-608):

“About the duration of the interventions, except for two out of them [35,76], the majority of the studies (six out of eight) informed about the impact of the exposition after some measurements points. Specifically, one of them [9] did not show positive effects in the post-treatment either in three weeks after the intervention. Likewise, another study [74] reported positive results six months after the intervention in mental and physical functioning of the women who attended the Education Groups, but not in those who participated in the Discussion Groups. In Study 4 [75], follow-up measures were performed after the implementation of the intervention. It was found better results in BI and self-esteem. At follow-up, only a significant improvement of large effect size in distress was observed. In two out of the studies [77,78], there were performed follow-up measures at one month, and at eight weeks, respectively, after the interventions. In the first of them, improvements over time in BI and self-esteem were not reported, and in the second one, it was reported better self-esteem levels at the eight weeks post-treatment. In another study [19] were carried out measures at post-treatment and 6 months’ follow-up after the end of the program. BI levels increased progressively from the first measure to the 6-month follow-up. However, self-esteem decreased post-treatment and it reported a new increase of its levels at the follow-up.

It is observed that in two studies [19,74] were performed follow-up measures six months after the treatment, as the Society of Prevention Research Criteria for Efficacy recommend [104]. Moreover, in Helgeson et al. [105], were exhibited distinct trajectories of change in psychological and physical adjustment to breast cancer over four years concerning the study 3 sample [74]. Likewise, follow-up measurements were carried out at different points in time (e.g. at 1, 2, and 3 months) in studies 2, 4, 6, and 7 [9,75,77,78]. In any case, studies that include some measure of follow-up already represent an advance, as this element is not always incorporated [75]. This leads us to point out the importance of conducting studies that explore the long-term impact of the intervention.”

Point 9. Line 116 change IB into BI

Response 9.

In line 116 (now line 96), the incorrect word IB has been changed to BI, as it has been ordered.

Point 10. In table 1 it is not clear why the authors have not used the same strategy of search in all databases? For instance, why not using the same starting date?

Response 10.

The used search strategy was the same in all databases. As is mentioned in LINE 165-166 of the revised manuscript: “All the databases were searched from inception to 8th October 2020”. In all of them, the next combination of Key Words and Boolean Operators: ("Breast Cancer" OR “Breast Cancer Women”) AND "Self-esteem" AND "Body image" AND ("Therapy" OR "Intervention") and language filters were included.

However, the cause why in Table 1 in “All dates” are different is because each Database's starting date was not the same (for instance, starting date in WOS, and SCOPUS are 1990 and 1997, respectively). Moreover, each database has its own characteristics. That is why in Table 1 appears different words before the descriptors and boolean operators, i.e., “topic”, “TITLE-ABS-KEY”, or no word if there was no option in the database.

Since there were no restrictions on the date when the search was carried out, the information about the period of time concerning databases is not significant. Therefore, it has been removed from Table 1 to facilitate the understanding.

Thank you for your question and to let us increase the improvement of the manuscript and its comprehension.

Point 11. The introduction is too long and there are several parts that are out of topic, such as lines 101 to 108. In relation to the lines, they should be presented in the entire manuscript. There is an error in the number of pages.

Response 11.

-Following the comments to the reviewer, the next paragraph from the Introduction section has been removed for being out of the topic: “This experience of BC takes place in a social context marked by three elements. Firstly, the social construction of cancer that emphasizes self-control and individual responsibility. At the same time, it marks a tension between cancer as a reality to be silenced and the emphasis on making the experience visible in terms of what is done to cure it, offering a selective and biased view of it. Secondly, the social and political meanings of the woman's breast that clash with each other and mark how they should be, felt and, used. For example, as a symbol of motherhood versus as a symbol of a constructed sexuality, putting the male gaze in the center and ignoring its consideration as an erogenous zone. Finally, gender roles for women. They are expected to build on the care of others and to set themselves up as an object of desire for men. However, these mandates clash with the experience of illness, which sometimes makes it difficult to provide care, an experience that is lived with suffering [42,43].” (lines 104-121 of the pre-revised manuscript).

References 42 and 43 have been also removed, and the remaining references have been adequately numbered after that change.

-The next sentence from the Introduction section has been also removed: “Similarly, it has been observed that in hormone-dependent types of cancer, hormone therapy acts by blocking the production or preventing the action of estrogens, affecting sexual desire and the ability to have orgasms [14,15].” (lines 65-69 of the pre-revised manuscript), for being out of the topic and to reduce the introduction section.

References 14 and 15 have been removed and the remaining references have been adequately numbered after that change.

-Following the reviewer’s comments, the number of pages has been changed and adequately numbered, and the lines have been presented in the whole manuscript.

Finally, we would like to acknowledge the important and useful suggestions and comments, making possible a better version of our work.

We hope all the changes fit well with your suggestions.

Yours sincerely,

Lucía Morales-Sánchez.

Reviewer 2 Report

Has the review protocol been registered in PROSPERO or other review protocol database? If yes, please add ID and other details. If not, please add this info in the manuscript.

inclusion/exclusion criteria should also contain the PICOS information and table. As an example please refer to table 1 of this manuscript doi: 10.3945/ajcn.113.069880

In a supplementary material authors should reported manuscript assessed in full but excluded with reasons, specifying the reason of exclusion. As an example please refer to https://www.ncbi.nlm.nih.gov/books/NBK262104/

Table 2 could be removed

In the results section, in addition to the paragraphs reported, authors should provide a general and harmonic overview of the results obtained bu the included studies (but avoiding to separately refer to each single study).

In discussion authors stated "These results could be influenced by sample size, professionals who apply the interventions, the meth-odology and instruments selected for the study, as well as the extent and duration of the intervention, and the covered contents. It is also recommended for future research to carry out randomized controlled trials to eliminate the possible bias in the effect of the interven-tions and to avoid limitations in the studies" I would suggest to consider the results obtained in a previous study where a comparison between self-reported measures and measurements performed by professional, in breast cancer women, was performed. In this study results shown no differences among the 2 methods. Please, refer to doi: 10.1016/j.clbc.2019.04.008

Author Response

Dear Reviewer 2,

Firstly, we would like to acknowledge your useful and relevant suggestions, which increases the quality of our manuscript. The responses to your comments are the following:

Point 1: Has the review protocol been registered in PROSPERO or another review protocol database? If yes, please add ID and other details. If not, please add this info to the manuscript.

Response 1:

The present systematic review has not been registered in PROSPERO or other review protocol database. This information will be reported in the manuscript as a limitation of the study, at the final of the discussion section: “Moreover, the present study has not been previously registered in any review protocol database, which may offer additional scientific information of the work(please, see lines 710-712).

Moreover, the suggestion to register it will be taken into account by the authors, as well as it will be remarkable in the acknowledgments. (please, see lines 772-775).

Point 2: Inclusion/exclusion criteria should also contain the PICOS information and table. As an example please refer to table 1 of this manuscript doi: 10.3945/ajcn.113.069880

Response 2:

In eligibility criteria section, authors have included the PICOS information and table (now TABLE 2) to provide more information about the research question and to facilitate the comprehension of the study, and the inclusion/exclusion criteria (Please, see Table 2, p.5).

It has been also added in lines 177-178 “The information concerning to participants, interventions, comparison, outcomes, and study design (PICOS) is showed in Table 2.”

Thank you for the provided example, it has been useful to perform our PICOS’s table.

Point 3: In a supplementary material authors should have reported manuscript assessed in full but excluded with reasons, specifying the reason of exclusion. As an example please refer to https://www.ncbi.nlm.nih.gov/books/NBK262104/

Response 3:

Following the reviewer’s suggestion, we have included in the Supplementary Materials, a table (Table S1) with the excluded articles (n=279) of the selection process, with the reasons for exclusions (please, see the ‘TABLE S1. List of excluded articles’ provided in an independent document). Thank you for the provided example, it has been taken into account to perform it.

In the results section, specifically at the subsection “3.1. Findings”, it has been added: “The list of excluded articles (n=279) and the reasons for exclusion are in Table S1 of the Supplementary Materials.” (please, see lines 221-222).

Moreover, it has been incorporated the “Supplementary materials” section in the Back matter, together with the Acknowledgments, Author Contributions, Conflicts of Interest, and References sections. (please, see lines 751-753): Supplementary Materials: The following is available online at http://...... Table S1: List of the excluded articles and reasons for exclusion.”

Point 4: Table 2 could be removed

Response 4:

The old Table 2 has been removed by following the referee’s suggestion. Now, Table 2 is referred to PICOS information. (Please, see Table 2, p.5).

Point 5: In the results section, in addition to the paragraphs reported, authors should provide a general and harmonic overview of the results obtained by the included studies (but avoiding to separately refer to each single study).

Response 5:

Following the reviewer’s suggestion, a paragraph was included in the results section. Specifically, in the subsection “Changes in BI and self-esteem after the interventions”, before the reported paragraphs, in order to provide a general overview of the efficacy of the interventions in body image and self-esteem outcomes among the breast cancer women after their application.

The added paragraph is the following: “Some of the studies found the interventions effective in terms of significant improvements in both self-esteem and BI variables after the implementation of the intervention [19, 74, 75, 78]. Other studies, however, only found changes in one of the variables, i.e., interventions produced positive changes in BI but not in self-esteem [35], or improved self-esteem but not BI [76]. Otherwise, some of the studies did not obtain significant results after the interventions, therefore they were not effective [9,77].” (please, see lines 287-294).

Point 6: In the discussion authors stated "These results could be influenced by sample size, professionals who apply the interventions, the methodology and instruments selected for the study, as well as the extent and duration of the intervention, and the covered contents. It is also recommended for future research to carry out randomized controlled trials to eliminate the possible bias in the effect of the interventions and to avoid limitations in the studies" I would suggest considering the results obtained in a previous study where a comparison between self-reported measures and measurements performed by professional, in breast cancer women, was performed. In this study results shown no differences among the 2 methods. Please, refer to doi: 10.1016/j.clbc.2019.04.008

Response 6:

Following the reviewer’s suggestion, it has been mentioned in that paragraph the study doi: 10.1016/j.clbc.2019.04.008 to incorporate its obtained results and to suggest to future research the need to analyze the measurements’ accuracy and the need to show the results obtained between the different applied methods.

The paragraph included is the following: “The included studies used self-reported measures, which could be a limitation in the obtained results. Therefore, it is suggested to analyze if the self-reported measurements are accurate. For instance, in Villarini et al. [94] it was demonstrated that self-report measures and measurements performed by professionals in breast cancer women showed no differences between the two abovementioned methods. This stands in contrast to Study 2 [9], in which the two methods obtained different results.” (please see lines 478-485).

Finally, we would like to acknowledge the important and useful suggestions and comments, making possible a better version of our work.

We hope all the changes fit well with your suggestions.

Yours sincerely,

Lucía Morales-Sánchez.

Round 2

Reviewer 1 Report

I do not have additional comments. 

Author Response

Dear Reviewer 1,

Newly, thank you very much for your useful suggestions and comments. It made possible the improvement of the review's quality and comprehension.

Yours sincerely,

Lucía Morales Sánchez.

Reviewer 2 Report

I would like to thank the authors for the efforts performed in improving the overall quality of the manuscript. 

I have only one clarification. When I previously asked the supplementary table with exclusion reason, I referred only to the articles assessed in full and then removed, so looking at the flow diagram I referred to the last 29 articles excluded. If possible, I would suggest modifying the supplementary table as suggested. 

In conclusion, according to the guidelines it is recommended to specify exclusion reasons only for the articles assessed in full.
As you can imagine, attaching a table with more than 200 articles is not useful/intelligible and readable. 

I'm satisfied with the other changes provided

Author Response

Dear Reviewer 2,

Thank you for your recognition for the work and our efforts. Newly, thank you very much for your suggestions and comments. It made possible the improvement of the review's quality and comprehension.

Concerning your suggestion, I have attached the document with the Supplementary Material (Table S1. List of articles excluded with reasons. n=29), according to your request. Moreover, in lines 220-221 of the resubmitted manuscript I have included the following sentence "The list of the last excluded articles (n=29) and the reasons for exclusion are in Table S1 of the Supplementary Materials.".

I am sorry for the confusion, but I thought you referred to the total articles excluded, except for those included in the review. I hope now the table fits with your request.

Thank you for your attention and support.

Yours sincerely,

Lucía Morales Sánchez.